# Plant Cuticles Exhibit Significant Mid-Infrared Emissivity in the Atmospheric Windows [note 1]

**DOI:** 10.3390/ijms26209917

**Published:** 2025-10-12

**Authors:** Antonio Heredia, Ana González-Moreno, José J. Benítez, Eva Domínguez

**Affiliations:** 1Departamento de Biología Molecular y Bioquímica, Instituto de Hortofruticultura Subtropical y Mediterránea “La Mayora”, Universidad de Málaga-Consejo Superior de Investigaciones Científicas, Universidad de Málaga, 29071 Málaga, Spain; gonzalezana@uma.es; 2Instituto de Ciencia de Materiales de Sevilla, Centro Mixto Consejo Superior de Investigaciones Científicas-Universidad de Sevilla, 41013 Sevilla, Spain; benitez@icmse.csic.es; 3Departamento de Mejora Genética y Biotecnología, Instituto de Hortofruticultura Subtropical y Mediterránea “La Mayora”, Universidad de Málaga-Consejo Superior de Investigaciones Científicas, Estación Experimental La Mayora, 29750 Málaga, Spain; edominguez@eelm.csic.es

**Keywords:** plant cuticle, mid-infrared, emissivity, reflectance, cutin, cell wall

## Abstract

As sessile organisms, plants have developed strategies to cope with exposure to high radiation. The plant cuticle is located at the interface between the plant and the surrounding environment, thus acting as a first barrier that protects plants against environmental conditions, including solar radiation. The isolated cuticles displayed notable absorptance in the infrared spectral range which, according to Kirchhoff’s law of thermal radiation, equals the emission dissipation ability. Comparison among the different cuticles showed that a significant range of their reflectance, transmittance, and absorbance spectra match the spectral regions known as atmospheric windows, between 3–4 and 8–13 microns, located within the mid-infrared region (MIR). They allow energy to pass through into the outer space. These optical parameters varied between cuticles from different plant species and they were not a simple function of the cuticle’s thickness but the product of its specific composition in combination with its molecular arrangement.

## 1. Introduction

The Earth’s atmosphere is a mixture of numerous chemical gases and compounds that lessens the dissipation of thermal radiation from the earth’s surface to the universe due to its high absorption and low transmittance [1,2]. However, the atmosphere is highly transparent to thermal radiation in some regions known as atmospheric windows (Figure 1). There are two main windows, between 3–4 and 8–13 microns, located within the mid-infrared region (MIR). Kirchhoff’s law of thermal radiation states that light absorbed by the surface of an object equals that emitted. Since radiation can either be reflected, transmitted or absorbed, determining how much light is reflected by a surface or is able to go through it is crucial to establishing its absorptance and hence emissivity. The reflectance of intact leaves within the MIR has been reported and, due to the lack of transmittance across the entire leaf, high emissivity values were measured [3,4,5].

The plant cuticle is a composite biomaterial that covers the epidermal surface of leaves, fruits, petals, and non-lignified stems [6]. It is composed of a biopolyester named cutin, remnants of cell wall polysaccharides, and waxes, which are a mixture of very-long-chain fatty acids, alkanes, and alcohols together with a small fraction of phenolics (for a cuticle scheme and their components, see Figure 1 of reference [7]. Cutin is the main component of a cuticle and constitutes the framework of a cuticle, whereas waxes and phenolics are present in low amounts and deposited and/or incorporated into the cutin matrix. Cuticle thickness is highly variable among and within species, organs and stages of development, ranging from 0.02 to over 20 microns; this can be translated to a specific weight that varies from 20 to over 3000 micrograms per square centimeter [7]. Cutin, waxes, and polysaccharides display moderate reflectance, thus conferring a variable transmittance and absorptance to the cuticle [8]. However, cuticle compounds absorb in the MIR spectral range due to multiple molecular vibrations [9]. Hence, the cuticle, due to its surface’s location, is the first level of plant interaction with radiation and could play an energetic role within the MIR. Cuticle deposition and biophysical characteristics have been reported to be impacted by increasing temperatures and drought [10,11,12,13]. Thus, completing our understanding of the energy balance of the electromagnetic light spectrum during its interaction with the plant cuticle is crucial for agriculture and plant ecosystems under current and predicted future climate conditions. Here, for the first time, we present a full characterization of reflectance, transmittance, and absorptance in the MIR range of isolated cuticles from different species and organs.

## 2. Results and Discussion

Reflectance spectra of isolated cuticles from fruits and leaves of different species displayed low values within the MIR, between 2 and 13 µm (Figure 2), in agreement with the results obtained by Richardson et al. [5] with intact leaves, which also showed similar spectral shapes. Total reflectance, diffuse and specular, was below 20% and close to 10% or lower within the 6–13 micron range in most species. No differences in reflectance between cuticles of leaves or fruits could be detected. Waxes, especially epicuticular ones which are located on the surface, have been traditionally reported to play a relevant role in reflectance [14]. However, the species studied have been reported to exhibit differences in major epicuticular wax compounds, from alkanes in the *Solanum lycopersicum* and *Capsicum annuum* fruits, to very-long-chain alcohols in *Hedera helix*, *Citrus aurantium* and *Agave americana* leaves and *Vitis vinifera* fruits, and a combination of very-long-chain alcohols and triterpenoids in *Olea europaea* fruits [15,16,17,18,19,20]. Still, the similarities in reflectance values and spectral shape among them indicate that there is no clear association between wax composition and MIR reflectance. It is noteworthy to mention that, in *S. lycopersicum* and *O. europaea* fruit cuticles, higher values of reflectance, close to 40%, were observed in the 2–3 micron range. This could not be related to specific wax components. Recent analyses of the optical properties of cuticles within the UV-Vis range have shown that cuticle reflectance is quite low in the UV region, similar to what was recorded here for the MIR, and only plays a more substantial role within the Vis region in a species-dependent manner [8,21]. Nevertheless, to ascertain the contribution of the wax layer to cuticle reflectance and assess the extent to which extent it could modify the calculated absorptance, optical properties were measured from the inner side of cuticles of two fruits and two leaves (Appendix A). Reflectance values were similar to those obtained from the outer side, thus indicating a very low level of participation by epicuticular waxes. Moreover, small differences in the cuticle outer surface’s topography were observed for the different species (Appendix A), yet they did not translate to notable changes in reflectance. These results underpin the notion that the role of waxes in reflectance may have been overemphasized and should not be generalized. The epicuticular contribution to reflectance in the UV region and MIR needs to be addressed in species displaying a variable morphology and composition of epicuticular waxes. Further work is necessary in this field.

Most of the MIR radiation was either transmitted or absorbed by the cuticle (Figure 2). The highest transmittance, over 60% along the whole spectrum, was observed for *C. aurantium* and *H. helix* leaf cuticles, which hence displayed the lowest average absorptance, around 25%. On the other hand, *S. lycopersicum* fruit cuticle had the lowest transmittance and highest absorptance, with an average of 70% within the MIR. The rest of the species displayed an intermediate behavior, with *A. americana* and *C. annuum* cuticles showing average transmittance values similar to those of absorptance. These results are in contrast with the very high emissivity, that is absorptance, that was reported for intact leaves and attributed to the cuticle [5]. These notably elevated values were consequences of the impossibility to determine cuticle transmittance in intact leaves, thus assuming all the radiation that was not reflected by the surface was indeed absorbed. Nevertheless, spectral analysis showed that most of the regions with highest cuticle absorptance coincided with atmospheric windows (Figure 3) and are present in the cuticles all the species studied. Indeed, cuticle absorptance within these windows is higher than the average of the overall MIR, ranging from 30 to 84%, depending on the species. The first window, very narrow, is located is between 3 and 4 microns and the second one between 8 and 13 microns. The chemical fingerprint associated with the first window is mainly assigned to the presence of hydroxyl functional groups and methylene chains, -CH_2_-, which are very abundant in the molecular structure of waxes and cutin of any plant cuticle [9]. The second spectral window, between 8 and 13 microns, is chemically more complex and includes some different vibrational modes of methylene chains (rocking, bending), ester bonds (C-O-C), C-O links, and aromatic rings. These vibrational modes are present in waxes, cutin, and phenolics and also include, with variable intensities, the modes assigned to the skeletal vibrations belonging to the structure of polysaccharides [9]. It should be remembered that the cuticle can be interpreted as a modification of the epidermal cell wall [6] and that the role of cellulose in IR emission and radiative cooling has been reported [22]. It is worth mentioning the notable absorptance peak around 9–10 microns observed in *A. americana*. This could be associated with the additional presence of cutan in this species, an aliphatic biopolymer composed of very-long-chain aliphatics linked by ester and ether bonds [6,9], since ether bonds show remarkable C-O-C vibrational modes in this region.

The cuticle also displayed a notable absorptance within the 5.5–6.5-micron region, outside the atmospheric windows (Figure 2 and Figure 3). Absorption within this region can be attributed to the carbonyl functional group (5.8 microns) present in the polyester cutin matrix and in some wax compounds, specifically in very-long-chain fatty acids, and to the vibration modes of double bonds and aromatic rings (6–6.5 microns) of phenolic compounds (Figure 3). This indicates that the cutin matrix, phenolic compounds, and some waxes can absorb and, therefore, emit, within and without the atmospheric windows. Thus, subtle compositional changes, for example, in the amount of free carboxylic acids, from waxes, phenolics, or a less esterified cutin matrix, would alter the contribution of these functional groups to emissions outside the atmospheric windows. Despite the fact changes in wax composition with latitude have been reported within species [23,24], it is unclear whether these changes would support this possibility.

The 6–6.5- and 12.5–13-micron spectral regions associated with phenolic compounds displayed very high emissivity in tomato fruit cuticle and to a lesser extent in the rest of the cuticles, with the exception of *C. aurantium* and *H. helix,* that showed little absorptance. The notable difference in emissivity observed between tomato and the other species within these regions could be related to the significantly higher phenolic amount that has been reported in the cuticle of red ripe tomato compared to the species studied here [8]. Indeed, a positive relationship between absorptance within either of these two regions and the amount of phenolics per cuticle surface area could be observed (Appendix A). Hence, this characteristic of cuticle phenolics could be added to the other functions that have already been ascribed to them such as acting as nanofillers conferring mechanical resistance, and providing photoprotection against UV radiation [8,21]. Nevertheless, it should be reminded that the first region (6–6.5 microns) falls outside the atmospheric windows. Thus, as it was mentioned above, phenolics will have dual emissions, within and without atmospheric windows.

Emissivity has been reported to proportionally depend on the thickness of the surface material [25]. Fruit pericarp and leaf cross-sections of the different species studied here show important variations in cuticle thickness (Appendix A). *C. aurantium* and *H. helix* had a notably thinner cuticle compared to the other species, as well as the highest transmittance and lowest absorptance, which seems to suggest that the above-mentioned relationship between thickness and emissivity can also be applied to the cuticle. However, *O. europaea* showed the highest cuticle thickness, approximately 20 microns, well above the other species, and yet the highest emissivity was observed for *S. lycopersicum,* which displayed a cuticle 4 times thinner than *O. europaea*. These results suggest that, despite the fact very low cuticle thickness could imply lower emissivity, it is not the only factor to consider. Indeed, cuticle composition, its molecular arrangement and density could also play a major role, especially in cuticles with intermediate to high thickness. In sense, it is worth mentioning that pore size has been reported to alter the emissive properties of a material [26], and given the porous nature of the cuticle [6,7], this is a trait that deserves further study.

As was mentioned above, polysaccharides contribute to emissivity within the 8-13-micron window. This is an important point since the non-cutinized region of the cell wall could also be significant in species with very thin cuticles, reinforcing the role of the cuticle. Additionally, the lignin-like domain identified in the inner part of the cuticle of some conifers [27] could also contribute to energy dissipation due to its aromatic domain, and the presence of C-C and C-O-C linkages.

## 3. Materials and Methods

### 3.1. Cuticle Extraction

Leaf and fruit cuticles were enzymatically isolated from different plant species following the protocol described in [8]. Pieces of leaves and fruits were incubated for at least 2 weeks in an aqueous solution of sodium citrate (50 mM, pH 3.7) with a mixture of fungal cellulase (0.2% *w*/*v* Sigma, St. Louis, MO, USA) and pectinase (2.0% *w*/*v* Sigma, USA) together with 1 mM NaN_3_ to prevent microbial growth. Once isolated from the epidermis, cuticles were incubated for another week in fresh enzymatic solution, and then rinsed in distilled water and stored under dry conditions. Mature fruits were employed to isolate cuticles from *Capsicum annuum* L., *Olea europaea* L., *Solanum lycopersicum* L., and *Vitis vinifera* L. and fully expanded leaves to isolate adaxial cuticles of *Citrus aurantium* L., *Hedera helix* L., and *Agave Americana* L. Cuticles were inspected under a stereomicroscope (Leica, Heidelberg, Germany) to avoid regions with cracks, surface defects, or stomata in the case of leaves.

Phenolics were estimated with a UV-Vis spectrophotometer after cutin depolymerization for 24 h in methanol with 1% KOH [8] and referred to the amount of cuticle per surface area determined, after weighing the flat pieces of cuticles from a known surface.

### 3.2. Tissue Sectioning and Microscopy

Pieces of fresh ripe fruits and fully expanded leaves were fixed in 4% paraformaldehyde for a few days, dehydrated in a series of ethanol, and embedded in a commercial resin (Leica Historesin Embedding Kit, Heidelberg, Germany). Samples were cross-sectioned into slices that were 4 µm thick using a Leica microtome (RM2125; Heidelberg, Germany). Sections were stained with Sudan IV to visualize the cuticle. Three biological samples per species were studied. Microphotographs were taken with a Nikon camera (DS-F) coupled to the microscope. For scanning electron microscopy, pieces of ripe fruits and fully expanded leaves were dehydrated, after fixation, in a series of ethanol and their surfaces inspected with a scanning electron microscope (JEOL JSM-6490LV, Tokyo, Japan).

### 3.3. Transmittance, Absorbance and Reflectance Measurements

Reflectance (%R) and transmittance (%T) spectra of plant cuticles were registered using a Jasco FT-IR 6800 (Jasco, Tokyo, Japan) spectrophotometer equipped with a 28° Michelson-type interferometer with corner cube mirrors, an integrated sphere with a gold-coated Lambertian surface, and an MCT detector (5000–500). Gold-coated masks of 4 mm diameter were used to hold the cuticles during measurements. Spectra were recorded after 150 scans at a 4 cm^−1^ resolution. The corresponding reflectance (R) and transmittance (T) spectra for each cuticle (in their outer and inner side) were recorded separately, putting the spectrophotometer in the appropriate technical configuration. Using the data obtained from these two spectra, the absorptance was calculated following the equation %Abs = 100 − (%R + %T), where Abs is absorptance and %R and %T are the registered and recorded data of reflectance and transmittance, respectively. Four to six cuticle samples from different plants were analyzed for each species.

## 4. Conclusions

From the above results, it can be concluded that the cuticle has a notable capacity to absorb and, hence, emit MIR radiation, but that this varies depending on the species. It should be mentioned that all cuticle compounds, cutin, polysaccharides, waxes and phenolics, participate, to various degrees, in this radiative mechanism. This could be envisioned as the ability of the cuticle to dissipate energy, not only specifically derived from MIR absorption, but also from UV and Vis radiation. In fact, most of the cuticle’s emissivity occurs within two atmospheric transparencies. Thus, the cuticle can be described from an optical point of view and, following the current technical terminology [1,2], as a combined multifunctional smart biomaterial with UV photoprotection, good absorbance capacity in the Vis spectral region and emission capacity in the MIR zone.

## Figures and Tables

**Figure 1 ijms-26-09917-f001:**
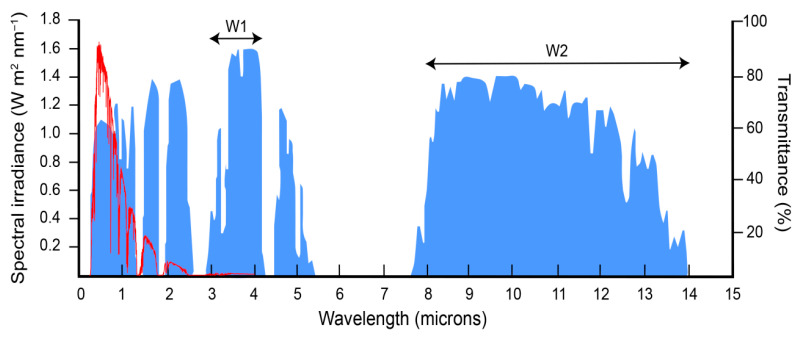
Spectral solar irradiance and percentage of atmospheric transmittance from UV to mid-infrared spectrum. The solar irradiance (AM 1.5 global standard) spectrum is shown in red, whereas atmospheric transmittance shown in blue. Atmospheric windows that allow radiation dissipation to the outer space correspond to high-transmittance regions; the two main windows (W1 and W2) are indicated. Solar spectrum data taken from https://www.pveducation.org/pvcdrom/appendices/standard-solar-spectra (accessed on 3 October 2025). Transmittance image modified from https://seos-project.eu/earthspectra/earthspectra-c04-p03.html (accessed on 3 October 2025).

**Figure 2 ijms-26-09917-f002:**
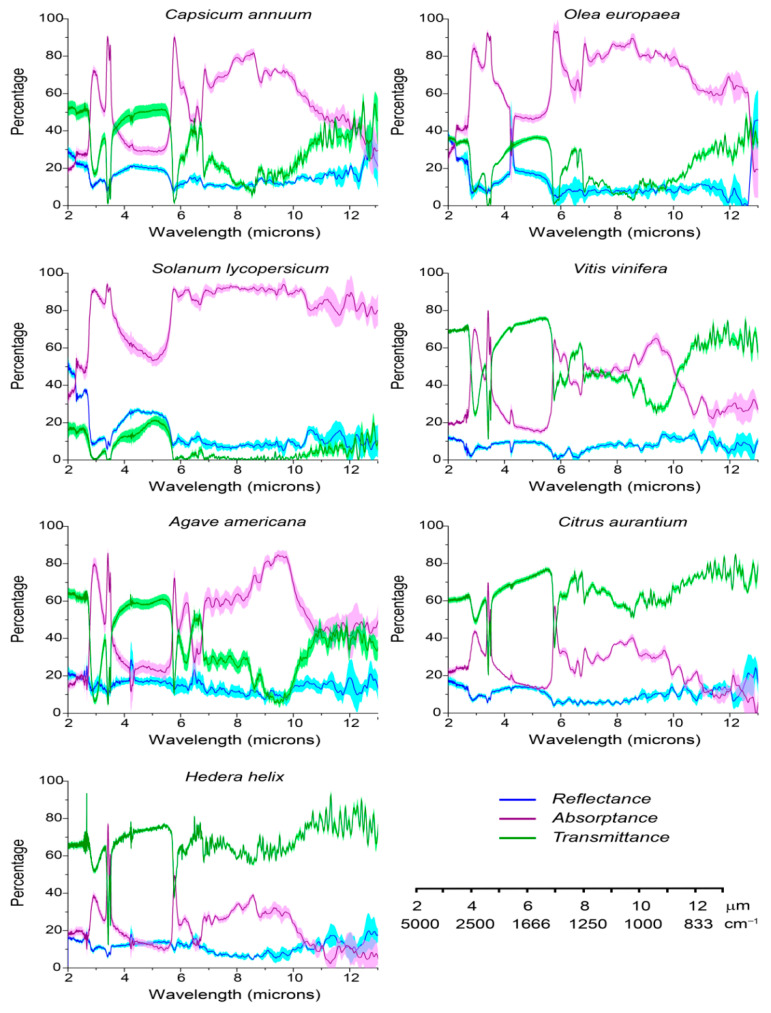
Transmittance, reflectance, and absorptance spectra of isolated cuticles from different species in the Middle Infrared region of the spectrum. *C. annuum*, *O. europaea*, *S. lycopersicum* and *V. vinifera* fruit cuticles. *A. americana*, *H. helix* and *C. aurantium* adaxial leaf cuticles. 4–6 biological replicates per species were studied. SE is given as shaded area. Wavelength scale conversion between microns and cm^−1^ is shown.

**Figure 3 ijms-26-09917-f003:**
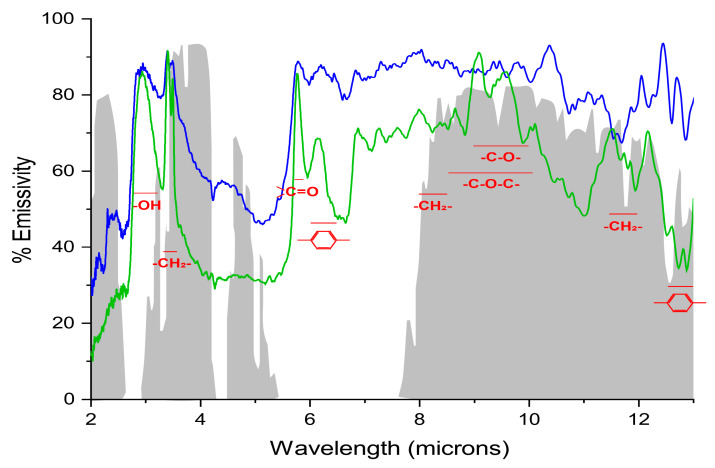
Identification of the main chemical bonds present in the plant cuticle whose vibrations are registered inside atmospheric windows. Solar transmission spectrum is shown in gray. Absorptance spectra of cuticles from *Solanum lycopersicum* red ripe fruit (blue) and *Agave americana* leaf (green) are also represented. Additionally, main functional group vibrations outside the atmospheric windows are also displayed.

## Data Availability

Data is contained within the article or Appendix A.

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
