# Peer review of "Plant Cuticles Exhibit Significant Mid-Infrared Emissivity in the Atmospheric Windows"

_ijms, 2025, doi:10.3390/ijms26209917_

Round 1

Reviewer 1 Report

Comments and Suggestions for Authors

This study investigates the emissivity properties of isolated plant cuticles in the mid-infrared (MIR) region. Using cuticles from different species and organs, the authors characterise reflectance, transmittance, and absorptance to assess their role in thermal energy dissipation. Results show that emissivity is not determined solely by cuticle thickness but depends strongly on composition and molecular arrangement. Importantly, high emissivity regions coincide with atmospheric windows, suggesting that plant cuticles act as multifunctional biomaterials contributing to radiative cooling and environmental adaptation.

The manuscript addresses an interesting and relatively unexplored aspect of plant cuticle function by linking their MIR emissivity with potential roles in radiative cooling. A major strength is the careful analysis of reflectance, transmittance, and absorptance, which clarifies misconceptions from previous intact-leaf studies. Overall, the contribution is original and provides a solid foundation for future investigations.

My recommendation is acceptance after minor revisions, which are listed below.

Lines 30-35: please include some appropriate references.

Line 38: Figure 1 appears as “stretched”: please provide an improved version of the image with a correct aspect ratio.

Lines 48-53: This paragraph should be rephrased, it is quite disorganised and not so easy to read.

Lines 54-55: Reference [6] states “… transmittance increased exponentially until reaching the maximum in the visible light range (400–800 nm), varying between 60% for A. americana and 90% for B. vulgaris.” when referring to isolated cuticles. On the contrary, in the present work the authors state that “Cutin, waxes and polysaccharides are transparent to the visible range of the electromagnetic spectrum, and display moderate reflectance…”. A trasmittance of 60% as in the case of A. americana does not indicate a complete transparency; moreover the experiments reported in reference [6] were conducted on the cuticles and not on the individual components (ie. cutin, waxes and polysaccharides). Lines 54-55 should be refined and the statement downsized.

Lines 100-101: Please provide a deeper explanation and contextualisation to this sentence.

Lines 112-114: Check the order and the numbering of the figures in the Supporting Information. Figure S3 and S4 appear in inverted order.

Line 224: the line should be properly formatted as an equation.

Author Response

Comment 1. Lines 30-35: please include some appropriate references.

Response: The revised version includes two references concerning this point.

Comment 2. Line 38: Figure 1 appears as “stretched”: please provide an improved version of the image with a correct aspect ratio.

Response: Fig. 1 appears "stretched" because for this kind of information (image, in this case) the x axis have to be longer that the corresponding y axis. A shorter or more "square" figure is not appropriate.

Comment 3. Lines 54-55: Reference [6] states “… transmittance increased exponentially until reaching the maximum in the visible light range (400–800 nm), varying between 60% for A. americana and 90% for B. vulgaris.” when referring to isolated cuticles. On the contrary, in the present work the authors state that “Cutin, waxes and polysaccharides are transparent to the visible range of the electromagnetic spectrum, and display moderate reflectance…”. A trasmittance of 60% as in the case of A. americana does not indicate a complete transparency; moreover the experiments reported in reference [6] were conducted on the cuticles and not on the individual components (ie. cutin, waxes and polysaccharides). Lines 54-55 should be refined and the statement downsized.

Response: Thanks for the comment. We have modified these sentences. 

Comment 4. Lines 100-101: Please provide a deeper explanation and contextualisation to this sentence.

Response: The sentence has been modified.

Comment 5. Lines 112-114: Check the order and the numbering of the figures in the Supporting Information. Figure S3 and S4 appear in inverted order.

Response: Thanks for the indication. The revised version has more figures in the main text and, in this sense, the numbering of the supporting figures has been modified.

Comment 6. Line 224: the line should be properly formatted as an equation.

Response: The equation is very simple and it has been incorporated in the text.

Reviewer 2 Report

Comments and Suggestions for Authors

The author analyzed the mid-infrared spectra of the cuticles of several different plants and conducted an analysis on them. There is a certain degree of novelty, but the following problems exist

1. The title in the text fails to convey the intended meaning and is not attractive enough.

2. The abstract does not present the main data and conclusions in the text.

3.In the Introduction section, the reason for conducting this research was not well explained, and there was no good introduction to each part of the stratum corneum. It is suggested that a diagram of the components of the cuticles could be added

4.In the Material&Methods section, there is no good explanation of the data collection steps, and the quantities of each variety are not detailed either. It could be made into a table. At the same time, I am also quite curious about how the spectral data was collected. Could a more detailed explanation be provided

5. Regarding the results, it is suggested that the supporting materials also be included in the text. Meanwhile, the author mentioned the inner and outter parts of the cuticles in the text. How to distinguish them? The text does not provide a good explanation of which spetra data belong to the outter and which belong to the inner

Author Response

Comments 1. The title in the text fails to convey the intended meaning and is not attractive enough. The title in the text fails to convey the intended meaning and is not attractive enough.

Response: We change the title: Emissivity spectra of isolated plant cuticles in the mid infrared spectral region.

Comments 2. The abstract does not present the main data and conclusions in the text.

Response: The revised version includes a modified and shorter abstract.

Comments 3. In the Introduction section, the reason for conducting this research was not well explained, and there was no good introduction to each part of the stratum corneum. It is suggested that a diagram of the components of the cuticles could be added.

Response: Thanks for the suggestion. If a new figure is added to the ms together some of the figures of the suplementary material  (as the referee indicates in his last comment) the final version of the work would have too many figures. We have modified the text indicating where exactly a general cuticle scheme can be found. On the other hand, we would like to indicate to the referee that our subject of study is the plant cuticle (and not the stratum corneum as the referee says), the membrane that covers the aerial parts of leaves, fruits and non-lignified stems of plant kingdom species. Concerning the objectives of our research we think that they are now clearly explained in the last new five lines before the Results and Discussion section of the ms.

Comment 4. In the Material & Methods section, there is no good explanation of the data collection steps, and the quantities of each variety are not detailed either. It could be made into a table. At the same time, I am also quite curious about how the spectral data was collected. Could a more detailed explanation be provided.

Response: The methodology used for the isolation of the different cuticles is described as is usual in the papers devoted to the chemistry, biophysics and biochemistry of plant cuticles (e.g. reference 8). In our opinion it is enough detailed. Concerning how were obtained the corresponding spectra we are modified the text.

Comment 5. Regarding the results, it is suggested that the supporting materials also be included in the text. Meanwhile, the author mentioned the inner and outter parts of the cuticles in the text. How to distinguish them? The text does not provide a good explanation of which spectra data belong to the outter and which belong to the inner.

Response: Thanks for your suggestion. Since to translate all the figures of the supporting material to the main text would increase too much the number of figures (this ms is being considered as Communication) we have decided include only one of them. At the same time we have modified the legend of the figure that the referee indicates concernig the outer and inner spectra of the isolated cuticles.

Round 2

Reviewer 2 Report

Comments and Suggestions for Authors

No modification in Abstract. The figures in the manuscript is not expressed clearly. As a communication, it's not attractive in novelty.

Author Response

Thanks for your comments. In the last version we have modified the Title, the Abstract explaining more details of the research and include a new sentence in the Introduction .  We have also modified one mistake concerning the reference about the scheme of a standard plant cuticle at the same time that the reference 9 is also included for this purpose.